# Tooth fracture frequency in gray wolves reflects prey availability

**Blaire Van Valkenburgh[1]\*, Rolf O Peterson[2], Douglas W Smith[3], Daniel R Stahler[3], John A Vucetich[2]**

[1]Department of Ecology and Evolutionary Biology, University of California, Los Angeles, Los Angeles, United States; [2]School of Forest Resources and Environmental Science, Michigan Technological University, Houghton, United States; [3]Yellowstone Center for Resources, National Park Service, Yellowstone National Park, Wyoming, United States

**Abstract** Exceptionally high rates of tooth fracture in large Pleistocene carnivorans imply intensified interspecific competition, given that tooth fracture rises with increased bone consumption, a behavior that likely occurs when prey are difficult to acquire. To assess the link between prey availability and dental attrition, we documented dental fracture rates over decades among three well-studied populations of extant gray wolves that differed in prey:predator ratio and levels of carcass utilization. When prey:predator ratios declined, kills were more fully consumed, and rates of tooth fracture more than doubled. This supports tooth fracture frequency as a relative measure of the difficulty of acquiring prey, and reveals a rapid response to diminished food levels in large carnivores despite risks of infection and reduced fitness due to dental injuries. More broadly, large carnivore tooth fracture frequency likely reflects energetic stress, an aspect of predator success that is challenging to quantify in wild populations.
DOI: https://doi.org/10.7554/eLife.48628.001

\*For correspondence:
bvanval@ucla.edu

**Competing interests:** The authors declare that no competing interests exist.

## Introduction

Competition for food among large carnivores is expected to increase when dietary overlap is high and the availability of preferred prey declines. Under such conditions, large carnivores are likely to spend more time feeding on their kills and consume them more completely, as well as scavenge more often. Greater carcass utilization often involves the consumption of less nutritious parts of a kill including bone, a behavior that increases tooth wear and the probability of tooth fracture, and might reflect higher levels of energetic stress. The association between tooth wear and bone consumption suggests that prey availability, carcass utilization, and tooth fracture are linked. When prey is difficult to kill, fewer prey are taken, and therefore carcasses might be more fully utilized with more bone consumed, and consequent increases in tooth fracture. If so, then rates of tooth fracture in large carnivores could be used as an index of prey availability in modern, historic, and ancient ecosystems.

Studies of tooth fracture frequencies in large Pleistocene carnivorans have revealed substantially higher rates of tooth breakage and wear in multiple species in both the Old and New Worlds. For example, the mean rate of tooth fracture on a per tooth basis is 2.3 ± 1.3% among 13 extant large (>21 kg) felids, canids, and hyaenids, whereas the same figure is 8.1 ± 3.5% for a sample of five large North American late Pleistocene felids and canids that represent distinct geographic locations (Alaska, California, Mexico, Peru) (*Van Valkenburgh, 2009*). Similarly, a study of temporal variation in dental wear and breakage in Pleistocene gray wolves of Great Britain found that per tooth fracture rates ranged from 2.5% to 8%, and noted that the higher rate was associated with lower prey diversity and the presence of a likely competitor, the brown bear (*Ursus arctos*) (*Flower and Schreve, 2014*). In both studies, the authors suggested that elevated tooth fracture frequencies in large

**eLife digest** Gray wolves roam many European and American landscapes, where they prey on large animals such as elk and moose. A healthy dentition is essential for these predators to kill, eat and defend themselves. As a result, they tend to avoid biting down on tough body parts, such as bones, so that their teeth do not break. If food becomes scarce however, the wolves may resort to consuming these hard elements, eating more of the carcasses and leading to more damaged teeth. It could therefore be possible to assess the food levels available to existing (or even extinct) wolf populations based on how many broken teeth the animals have. However, older individuals are also more likely to have more damaged teeth, so age would need to be taken into consideration.

Van Valkenburgh et al. decided to evaluate whether it was indeed possible to deduce how much food was available to groups of wolves based on teeth damage. Tooth wear and fracture were quantified in three current populations of gray wolves whose skulls had been collected and preserved in natural history collections. For each group, there were data available about the variations of number of moose per wolf over time, and how much of the carcasses the wolves were consuming. The analyses showed that indeed, when prey became less abundant, the wolves ate more of the remains – including the bones – and therefore broke more teeth.

These conclusions can be applied to other large predators and even to extinct species such as dire wolves or sabertooth cats. Tapping into the potential of museum specimens could help to retrace environmental conditions and the history of animals now long gone.
DOI: https://doi.org/10.7554/eLife.48628.002

Pleistocene carnivorans reflect intensified competition for kills and consequent increases in heavy carcass utilization and scavenging.

However, the inference of an association between prey availability, competition, increased bone consumption and higher tooth fracture rates is complicated by several factors. First, because the probability of having at least one broken tooth increases with age (*Van Valkenburgh, 2009*), populations dominated by older individuals are likely to have higher rates of tooth fracture and heavier tooth wear than those dominated by younger individuals. Most museum collections of skulls do not have associated age data so it is difficult to control for this potential bias. Second, museum collections almost never have data on levels of prey availability or carcass consumption behavior for the sampled predators. Third, the most commonly broken teeth are canines, and these teeth have multiple functions as weapons in combat and predation as well as in feeding. Consequently, increases in canine tooth breakage are more problematic to interpret than increases in premolar or molar breakage.

To better understand the causes of variation in tooth fracture frequency among fossil and living large carnivorans, we collected dental wear and fracture data from gray wolves (*Canis lupus*) representing three well-studied populations that had associated data on prey availability, and in some cases the degree of carcass consumption and age of death for each wolf. These include samples of wolves from Isle Royale National Park (USA), Yellowstone National Park (USA) and Scandinavia. In both Isle Royale and Scandinavia, greater than 90% of the prey of wolves is moose (*Alces alces*), but the mean ratio of moose to wolves is 499:1 in Scandinavia and only 55:1 in Isle Royale (*Sand et al., 2012*). Kill rates (#kills/wolf/day) are three times greater in Scandinavia than Isle Royale (*Sand et al., 2012*), and in association with this, Scandinavian wolves tend to consume less of their kills (about 70%) than Isle Royale wolves (about 90%) (*Vucetich et al., 2012*). Consequently, wolf tooth wear and fracture rates are expected to be greater in Isle Royale than Scandinavia. In Yellowstone (YNP), over 90% of the prey are elk (*Cervus canadensis*) (*Metz et al., 2012*), and the ratio of elk to wolves has declined sharply from over 600:1 to around 100:1 since the initial reintroduction of wolves in 1995 (*Figure 1*) . As the availability of elk declined in Yellowstone over the last 20 years, it is possible that carcass consumption levels increased, as well as rates of tooth wear and fracture. Because age at death was recorded for most of the preserved Yellowstone wolves, we can compare tooth wear and fracture frequency between wolves of similar age during times of prey abundance and times of relative prey scarcity. The three wolf populations, Isle Royale, Scandinavia, and Yellowstone provide windows into three different but overlapping scenarios of prey availability with consequent

implications for dental attrition, one with abundant prey (Scandinavia), a second with more limited prey (Isle Royale), and the special case of Yellowstone in which there is the potential to track predator tooth damage alongside a decline in prey numbers. The tooth wear and fracture data from these three populations were also compared with data collected previously from historic populations of 223 North American gray wolf skulls of known provenance but without associated data on prey availability or feeding behavior.

Previous work on tooth wear and fracture in mammals has focused on the relationship between diet and dental attrition. Early experimental studies on laboratory populations of rodents (*Carlsson et al., 1966*), primates (*Teaford and Oyen, 1989*), and carnivorans (*Berkovitz and Poole, 1977*) established the fact that abrasive diets that include grit or other tough materials increase rates of tooth wear relative to diets composed of relatively soft foods. Studies of wild populations are rare, but a recent comparison of tooth wear and fracture in wild coyotes relative to a matched age sample of captive-reared coyotes fed a relatively soft diet lacking any bones found more rapid rates of wear and higher tooth fracture in the wild sample (*Curtis et al., 2018*). The authors suggested that this likely reflected a greater inclusion of bone in the wild coyotes' diets, but behavioral data to confirm this were not available. In all these studies, the primary question has been the association between dental attrition and diet, with little or no consideration of how tooth wear might provide a window into levels of food stress or competition. Our comparison of dental attrition and carcass utilization in three wild populations of gray wolves expands the application of tooth fracture analysis to larger ecological questions, such as variation in the intensity of intra- and interspecific competition within carnivore guilds.

## Results

### Overall fracture frequency

The percentage of individuals with at least one tooth broken in life averaged 51% across all five samples, but ranged from less than 38% in the Scandinavian and early Yellowstone (1995–2006) populations to 72% in Isle Royale wolves (*Table 1*). The Isle Royale and later Yellowstone samples did not differ significantly from one another in the fraction of individuals with at least one broken tooth, 72% and 64%, respectively (p=0.3). Both of these samples exhibited significantly more individuals with broken teeth than either of the samples with known high prey:predator ratios, Scandinavia and early Yellowstone (p<0.001). Notably, the early and later Yellowstone samples differed significantly in the percentage of individuals with at least one broken tooth with and without including canine teeth (p=0.001); the fraction was 38% with canines (35% without) between 1996 and 2006, whereas in the following decade it rose to 64% with canines (63% without). It should be noted that the number of broken teeth per individual in the later Yellowstone sample is significantly greater than in the earlier sample but still usually represents less than ten percent of the entire tooth row (e.g. less than five of the 42 teeth per individual).

Results were similar for fracture frequency on a per tooth basis. Isle Royale wolves displayed the highest fracture frequency (8.6%) and Scandinavian and early Yellowstone wolves the lowest, 1.7% and 1.8%, respectively (*Table 1*). Although the fracture rates per individual were similar between Isle Royale and later Yellowstone wolves, Isle Royale wolves exhibit significantly more tooth fracture on a per tooth basis, 8.6% as opposed to 4.6% in later Yellowstone wolves (p<0.001). As was the case for breakage per individual, later Yellowstone wolves significantly exceeded early Yellowstone wolves in fracture frequency (4.6% as opposed to 1.8%; p<0.001) when canines were included and also when only incisors and cheek teeth were considered (3.9% vs. 1.3%; p<0.001), thus removing any bias due to a prevalence of canine tooth fracture. Within each of the five samples, tooth fracture frequencies calculated with and without canine teeth were very similar and did not differ significantly (p=0.1–0.5, *Table 1*).

### Tooth fracture by position

The differences in fracture frequency among the five samples were not due to higher fracture frequency at any single tooth position such as the canines. Instead, elevated fracture frequencies usually occurred across the entire tooth row. For example, the two samples with the highest fracture frequencies, Isle Royale and later Yellowstone, broke all their teeth, incisors, canines, premolars,

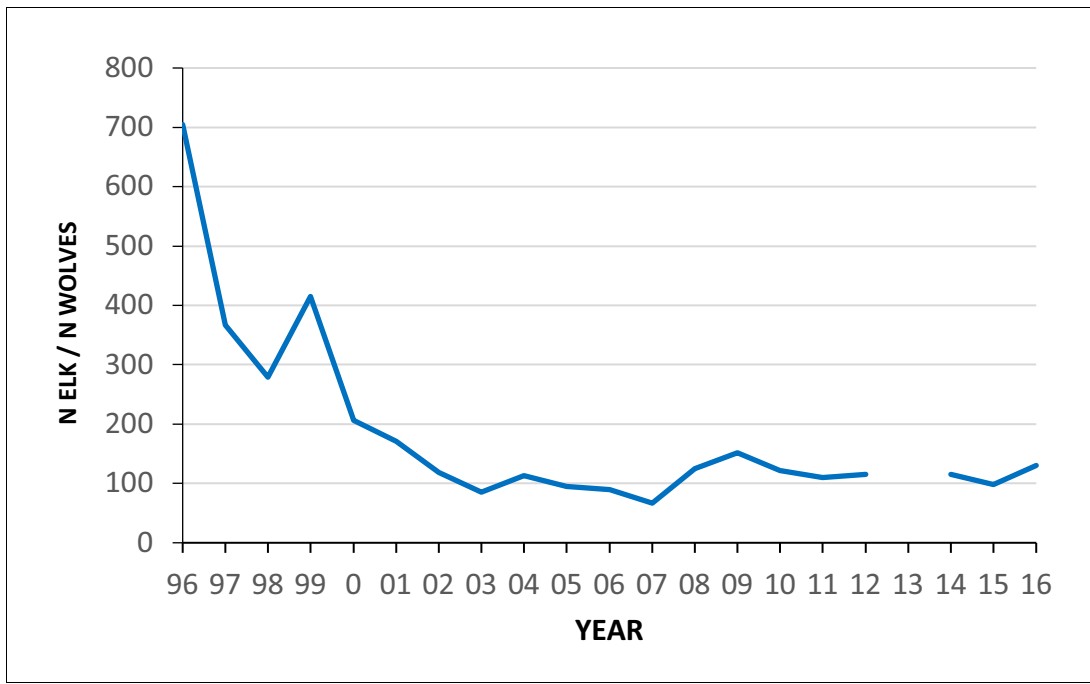

**Figure 1.** Estimated ratio of the number of elk (all ages, both sexes) to wolves in the northern range of Yellowstone National Park between 1996 and 2016. Data from Northern Yellowstone Cooperative Wildlife Working Group and Yellowstone National Park.
DOI: https://doi.org/10.7554/eLife.48628.010

carnassials and post-carnassial molars, more often than was observed in the remaining three samples (*Figure 2*, top; Supporting Information *Supplementary file 1* Table S2). Differences between these two samples and the remaining three were significant ($p<0.05$) at all tooth positions except the canines in the case of the two Yellowstone samples. Notably, the two Yellowstone samples differed most in the fracture frequencies of all teeth except the canines. Whereas canine tooth fracture rate increased approximately 50% between the first and second decade, fracture rates for incisors, premolars and carnassials more than tripled (*Figure 2*, top; Supporting Information *Supplementary file 1* Table S2). Relative to both Scandinavian and other North American wolves, early Yellowstone

**Table 1.** List of gray wolf skull samples indicating the number of individual skulls examined for each (N skulls), the total number of teeth (N teeth), the percent of individuals with at least one tooth broken, the percent of total teeth broken both with canine teeth included (% broken teeth) and excluded.
As noted in the text, fracture frequencies with and without canine teeth included did not differ significantly for any of the five samples.

| Sample | N (skulls) | N (teeth) | % individuals w/≥ 1 brkn tooth | % broken teeth | % broken teeth with canine teeth excluded |
|---|---|---|---|---|---|
| Isle Royale NP | 64 | 1866 | 71.9 | 8.6 | 7.4 |
| Scandinavia | 94 | 3778 | 33 | 1.7 | 1.5 |
| Other NA wolves* | 223 | 8619 | 47.1 | 3.4 | 3.1 |
| YNP, 1996–2006 | 77 | 2991 | 37.7 | 1.8 | 1.3 |
| YNP, 2007–2016 | 83 | 3237 | 64 | 4.6 | 3.9 |

*These include individuals from Alaska (n-74), Canada and Idaho (n = 66), and New Mexico and Texas (n = 83) For fracture frequency data for each of these, see *Supplementary file 1*, Table S1.
DOI: https://doi.org/10.7554/eLife.48628.003

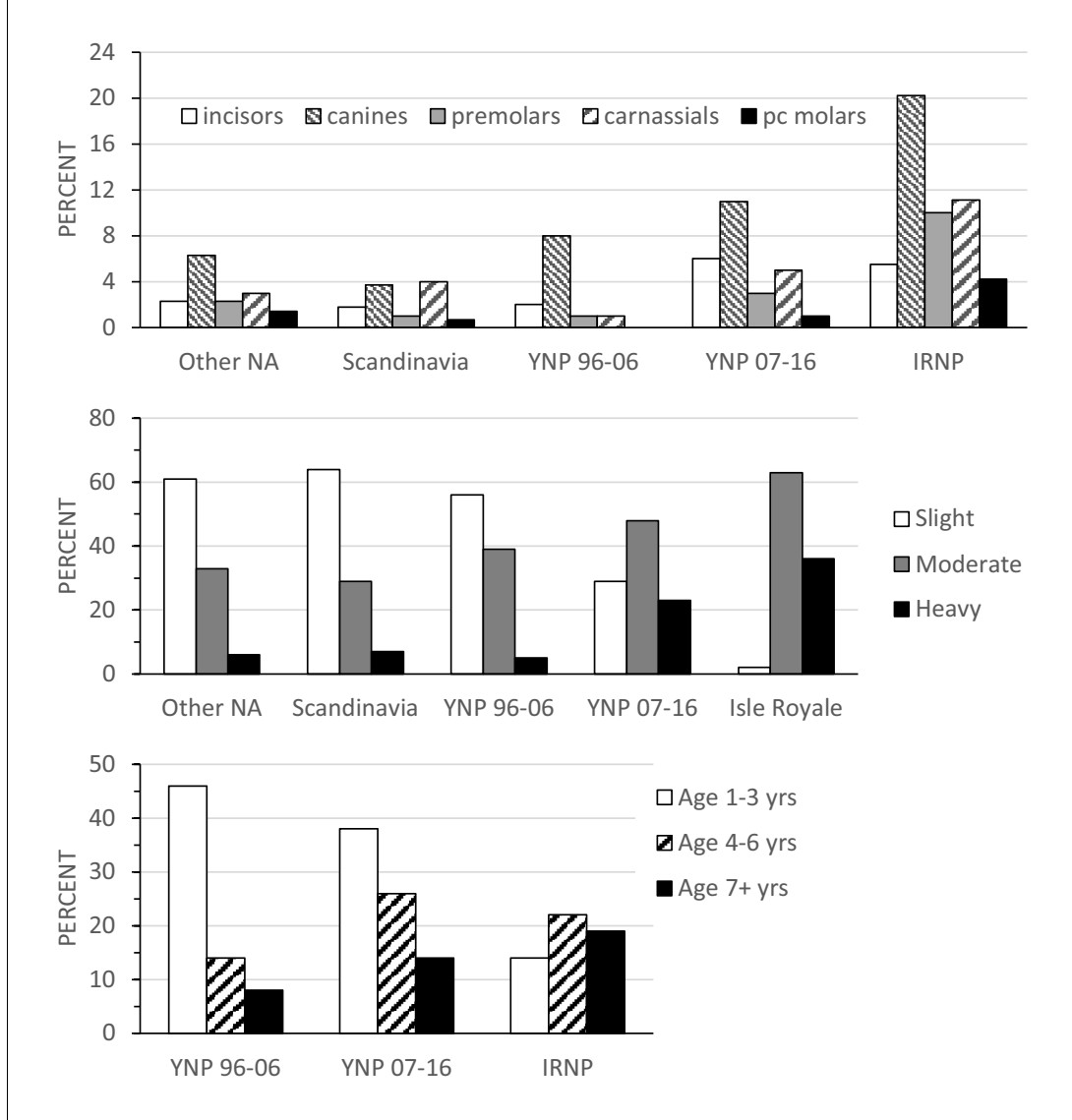

**Figure 2.** with one supplement. (Top) Percent of teeth broken for each tooth type within each sample. For tooth sample sizes see Supporting Information *Supplementary file 1* Table S1. (Middle) Percentage of individuals assigned to each wear stage class within each sample. (Bottom) Percent of individuals in each age class (years) for the three samples with age data, YNP 1996–2006, YNP 2007–2016, and Isle Royale NP.

DOI: https://doi.org/10.7554/eLife.48628.004

The following figure supplement is available for figure 2:

**Figure supplement 1.** Examples of gray wolf mandibles showing slight (top), moderate (middle) and heavy (bottom) tooth wear.

DOI: https://doi.org/10.7554/eLife.48628.005

wolves were unusual in exhibiting relatively high canine tooth fracture incidence along with low carnassial and post-carnassial molar fracture incidence.

## Tooth fracture frequency vs. wear stage and age at death

The high fracture frequencies for Isle Royale and later Yellowstone wolves are associated with a greater percentage of individuals with moderate and heavy tooth wear (*Figure 2*, middle; Supporting Information *Supplementary file 1* Table S2). Over half (56%–64%) of the Scandinavian, early Yellowstone and other North American wolf samples, respectively, were classified as slight wear. By contrast, only 2% (1/64) of Isle Royale individuals fell into the slight category, and just 29% (24/83) of the later Yellowstone fell into the slight category (Supporting Information *Supplementary file 1*

Table S2). Because wear stage is positively correlated with an individual's age, the preponderance of moderate and heavily worn individuals and associated higher tooth fracture rates within Isle Royale and the later Yellowstone samples might reflect a bias toward older individuals in these two samples rather than differences in diet or feeding behavior. Fortunately, it is possible to control for this potential bias because both the Yellowstone and Isle Royale samples have associated age data.

Analysis of the age distributions for the two Yellowstone and Isle Royale samples does reveal significant differences in the predicted direction. Both the Isle Royale and later Yellowstone sample have a greater proportion of older (>4 years) individuals than the early Yellowstone sample (p<0.001; *Figure 2*, bottom). Nevertheless, when rates of tooth wear are compared within similar age classes (1–3 years, 4–6 years, and 7+ years), it is apparent that both the Isle Royale and later Yellowstone samples wore their teeth more rapidly as they aged (*Figure 3*). For example, the early Yellowstone sample had individuals with slight wear in all three age groups, whereas the later Yellowstone and Isle Royale groups had almost no slightly worn individuals among 4–6 year-old wolves, and none at all after age seven. Instead, the proportion of individuals assigned to the heavy wear class was greater in both of these samples in the two later age classes (4–6 years, 7+ years).

Tooth fracture rates also differed by age at death among the three samples with or without canine teeth included (*Figure 4*). The Isle Royale wolves have significantly more fractured teeth at any age relative to both groups of Yellowstone wolves. Between the two Yellowstone samples, the 2007–2016 sample consistently exhibited higher fracture frequencies. The difference approaches significance in the 4–6 year-old age class (p=0.08) and is highly significant (p<0.01) for the 7+ year-old age class. Whereas in the later Yellowstone sample, 10% (66/671) of the teeth in individuals older than 6 years were broken, the same figure was only 3% (16/562) for the earlier Yellowstone wolves.

## Skeletal utilization

Between 1996 and 2016, necropsies were performed on 2372 adult elk largely from Yellowstone's northern range representing a total of 99,624 possible skeletal elements (excluding the pelvis and cranium) as categorized by the observers (*Supplementary file 1*, Tables S3, S4). Of these elements, approximately 17% were missing and presumed removed by wolves. Forelimb elements (scapula, humerus, radius + ulna, metacarpus) appear often to have been removed together and were more likely to be taken than hindlimb elements (*Supplementary file 1*, Table S3). This is not surprising given that the hindlimb has a bony articulation with the inominate (pelvis) bone, whereas the forelimb has a muscular attachment to the ribcage that is more easily severed. Like the hindlimb, vertebrae and dentary bones are more difficult to remove given their ligamentous and bony articulations and they were more likely to remain at kill sites on average than forelimb elements.

The proportion of the skeleton that was removed varied over time, ranging from a low of 7% in 1997 to a high of 35% in 2016 (*Figure 5*, *Supplementary file 1*, Table S4). Changes over time in the overall proportion of the skeleton removed were mirrored by those of the humerus alone, highlighting the apparent preference for the forelimb mentioned above. There is considerable scatter in the plot of skeletal utilization over time (*Figure 5*) with more variance in the first decade than the second (i.e., heteroscedasticity). Consequently, the data are not appropriate for linear regression and a three-year simple moving average was estimated. There is the suggestion of an upward trend in the last decade, indicating that Yellowstone wolves processed carcasses more fully in the second decade after elk numbers declined. When the necropsy data are split into two chronological samples, 1997–2006 and 2007–2016, the proportion of skeletal elements removed is 15% for the early sample and significantly less than that for the later sample, 20% (p=0.001).

## Discussion

In the sampled wolves, a rise in tooth fracture frequency is associated with a decline in the availability of their primary prey, as estimated by the ratio of prey to predator abundance. The two samples from areas with high prey to predator ratios, Scandinavia and early Yellowstone, are similar in exhibiting low tooth fracture frequencies, both on a per tooth and per skull basis. In contrast, the two samples from areas with much lower prey to predator ratios, Isle Royale and later Yellowstone, have greater rates of tooth wear and fracture, even if canine teeth are excluded and differences in age distribution are considered. Although the per tooth fracture frequency in later Yellowstone (4.6%) is

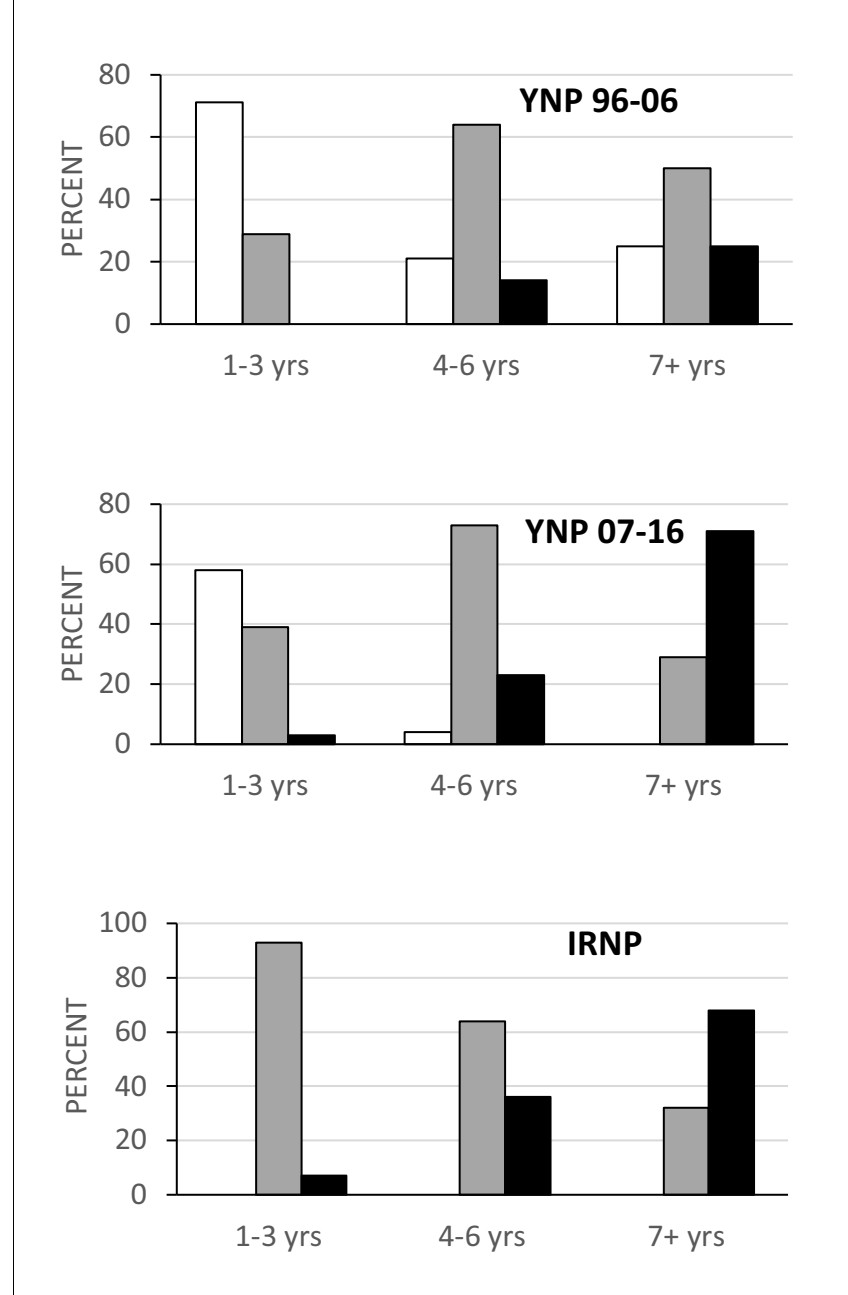

**Figure 3.** Proportion of individuals in each wear class grouped by age for the three samples with age data. Wear class: slight, white; moderate, gray; heavy, black. Note the persistence of individuals in the slight wear class as they age in the early Yellowstone sample relative to the other two samples.
DOI: https://doi.org/10.7554/eLife.48628.006

not as high as that of Isle Royale (8.6%), it is significantly larger than the fracture frequency for early Yellowstone wolves (1.8%), suggesting a change in feeding behavior.

The change in feeding behavior that is most likely to have caused the rise in tooth fracture is increased bone consumption. This is supported by previous interspecific comparisons of carnivoran tooth fracture and diet (*Van Valkenburgh, 1988*; *Van Valkenburgh, 2009*; *Mann et al., 2017*), as well as by the distribution of fracture across the tooth row in the wolves sampled for this paper. As has been shown in previous studies of tooth fracture, the canine teeth are the most likely to be broken, probably due to their elongate shape and use in combat and killing prey. Nevertheless, teeth

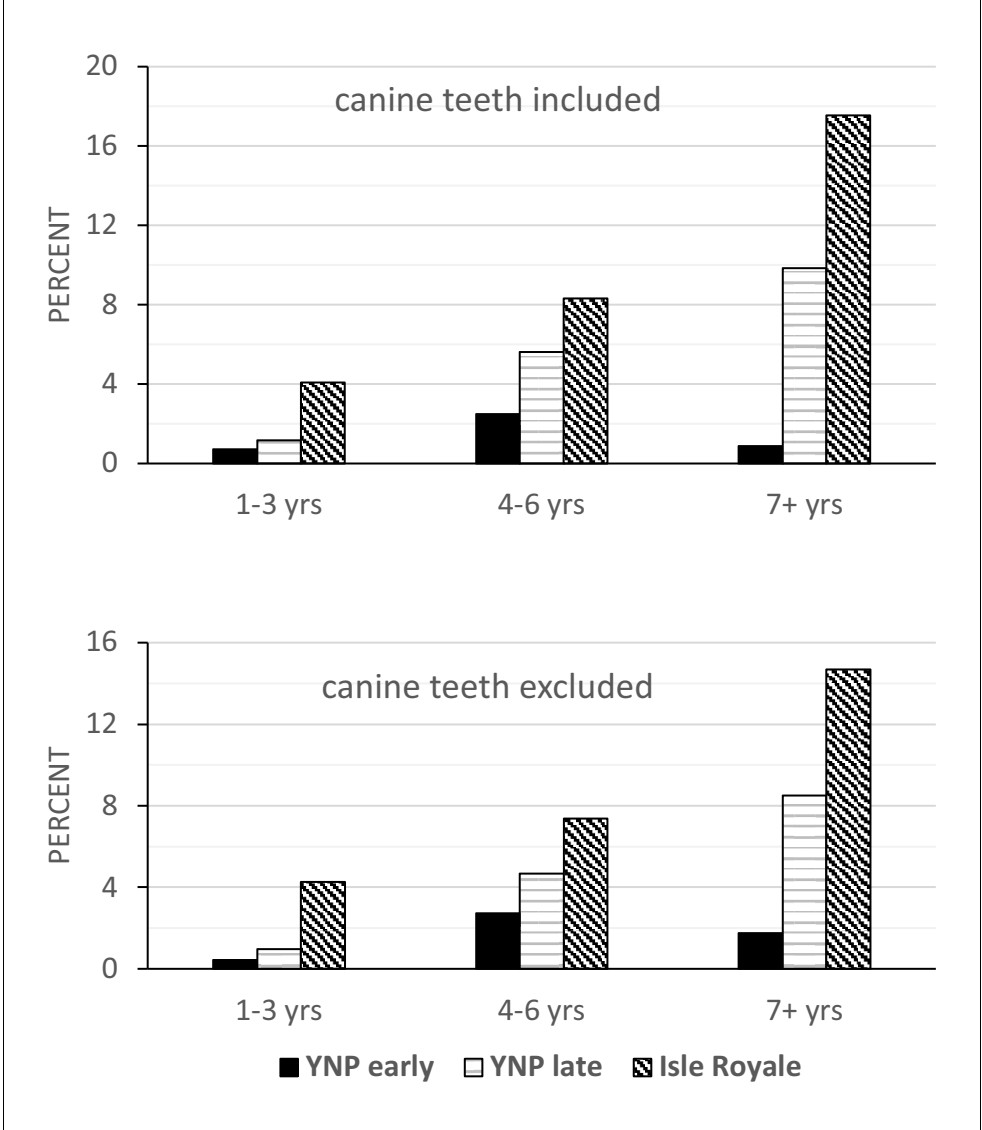

**Figure 4.** with one supplement. Tooth fracture frequency on a per tooth basis relative to age class (in years) for Yellowstone 96–06, Yellowstone 07–16, and Isle Royale wolves, with canine teeth included (top) and excluded (bottom).

DOI: https://doi.org/10.7554/eLife.48628.007

The following figure supplement is available for figure 4:

**Figure supplement 1.** Scatter plot with separate linear regressions of the number of broken teeth relative to the age of an individual for the three samples with known-age wolves: blue, YNP1, Yellowstone 1997–2006; red, Yellowstone wolves, 2007–2016; green, Isle Royale National Park wolves.

DOI: https://doi.org/10.7554/eLife.48628.008

involved in gnawing (incisors) and food processing (premolars and molars) are broken more often in Isle Royale and later Yellowstone samples than in the early Yellowstone, Scandinavian, or other North American samples. Notably, the two Yellowstone samples differ much less in canine tooth fracture frequency than carnassial tooth fracture frequency. Whereas canine tooth breakage rose by 50% between the early and later cohorts, carnassial tooth breakage went up 500%. Additionally, *Cubaynes et al. (2014)* found density-dependent rates of aggression in Yellowstone's northern range wolves, with significantly lower rates of aggression corresponding with later Yellowstone samples showing greater canine tooth breakage. Furthermore, the proportion of winter biomass acquired by later Yellowstone wolves from bison (*Bison bison*) increased significantly (*Metz et al.,*

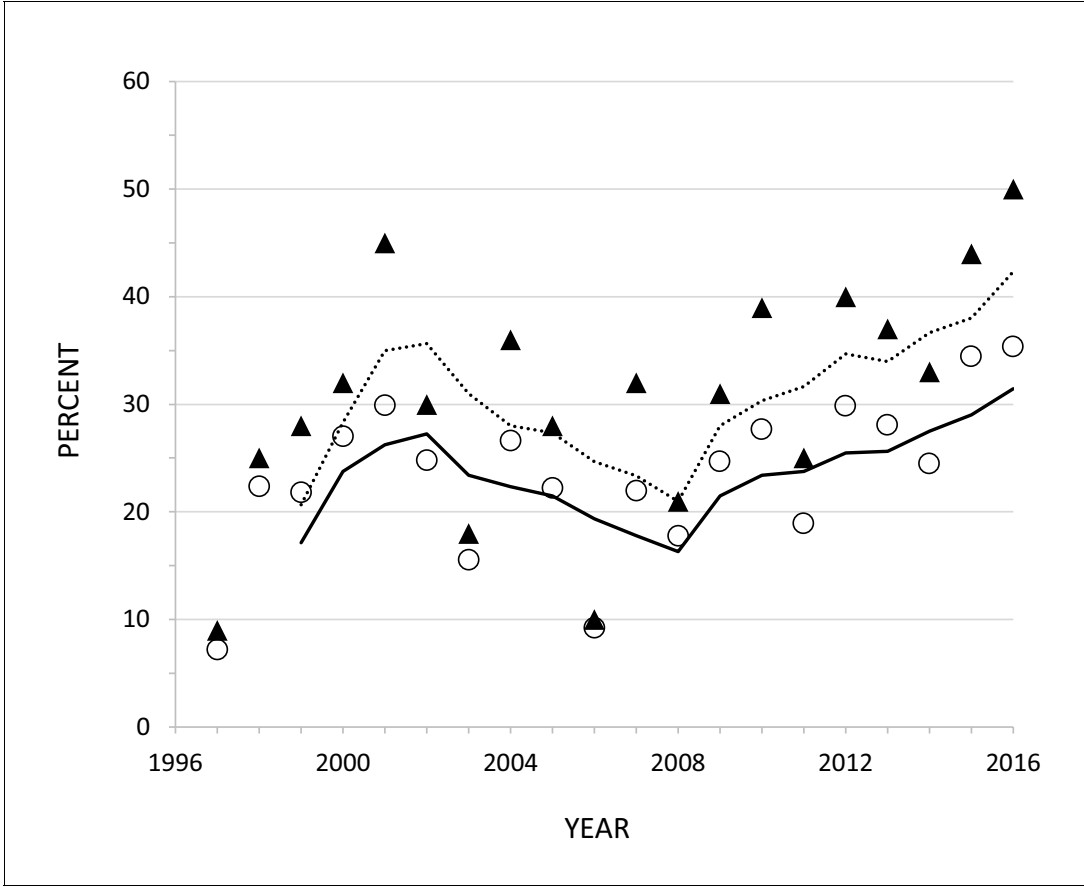

**Figure 5.** Proportion of all skeletal elements (open circles, solid line) and humeri only (triangles, dashed line) removed each year based on field necropsies of 2732 kills of adult elk in YNP between 1997 and 2016. The lines represent 3 year simple moving averages. Supplemental Information Figure Legends.

DOI: https://doi.org/10.7554/eLife.48628.009

*2016*), largely from scavenging winter-killed carcasses. Consequently, utilization of bison carcasses with larger, thicker bones may explain some of the increased frequency in tooth fracture. These patterns reinforce the idea that it is a shift in feeding behavior rather than one in levels of aggression or a change in predatory methods that is responsible for the bump up in tooth breakage.

The distribution of individuals by tooth wear stage also differed between the high and low prey to predator samples in ways that are consistent with more bone consumption in the latter. There were greater proportions of individuals with slight wear in the Scandinavian, early Yellowstone, and other North American samples relative to both Isle Royale and later Yellowstone, in which individuals with moderate to heavy wear predominated. These differences might reflect differences among the samples in age distribution rather than feeding behavior, given that older individuals have more heavily worn teeth, but this was not the case. Age data were available for both the Yellowstone and Isle Royale samples, and when individuals of similar age are compared, it is clear that later Yellowstone and Isle Royale wolves wore their teeth more rapidly as they aged and were more likely to fracture their teeth at any age than early Yellowstone wolves (*Figures 3* and *4*).

Even though both Isle Royale and later Yellowstone wolves exhibit elevated rates of tooth wear and fracture relative to the other wolf samples, the Isle Royale wolves have much higher rates of fracture on a per tooth basis than the later Yellowstone wolves. On a per tooth basis, the frequency of tooth fracture for Isle Royale wolves is similar to that of sampled late Pleistocene gray wolves and dire wolves. Isle Royale wolf per-tooth fracture rates are nearly double that observed for later Yellowstone wolves and about three to four times the rates observed in all the other extant samples. This may reflect more extreme food limitation in Isle Royale, but there is another factor that might

be relevant. Isle Royale wolves are the most inbred of the three focal samples, the other two being Yellowstone and Scandinavia. Previous work on congenital malformities in both the highly inbred Isle Royale wolves and moderately inbred Scandinavian population found a significantly greater incidence of skeletal malformations in the former (*Räikkönen et al., 2009*; *Räikkönen et al., 2013*). Perhaps inbreeding depression has affected tooth strength in Isle Royale wolves, although there are no other studies of inbred mammals that document impacts on tooth strength or breakage. Alternatively, perhaps the higher fracture frequencies in Isle Royale are due to a greater size difference between predator and prey. Both Isle Royale and Scandinavian wolves kill moose, but the Scandinavian wolves are significantly larger (*Sand et al., 2012*). Given their smaller size, Isle Royale wolves may have more difficulty killing adult moose and therefore need to consume kills more completely. Moreover, although wolves prefer to kill calves, the number of available calves each year is much less in Isle Royale than Scandinavia and consequently Isle Royale wolves kill a greater proportion of adults (*Sand et al., 2012*). Finally, it is possible that the probability of breaking a tooth increases with the number of broken teeth. The presence of broken teeth might lead to malocclusion or other abnormalities that then increase the probability of heavier wear and/or tooth breakage, producing more rapid increases in tooth fracture rates and wear as individuals age. Isle Royale wolves may be further along this trajectory than the later Yellowstone wolves.

The data presented here support the idea that rates of cheek tooth fracture in large carnivores can be used as an index of prey availability in modern and ancient ecosystems. Consequently, the elevated rates of tooth fracture in large Pleistocene carnivores might be interpreted as evidence of relatively high predator to prey ratios and greater top-down forcing on large herbivore populations than are typically observed today (*Ripple and Van Valkenburgh, 2010*; *Van Valkenburgh et al., 2016*). However, 'prey availability', defined as the relative ease of acquiring and consuming a kill, can be more complicated than the metric used here, the numbers of prey relative to predators. For example, in the case of wolves, prey are easier to catch in severe winters with deep snow than milder winters (*Mech et al., 2001*), as well as during late winter versus early winter when prey are in better nutritional condition (*Metz et al., 2012*). In addition, prey availability can be affected by interference as well as exploitative competition. Intraguild competition is relatively intense within guilds of large mammalian carnivores, and manifests itself in multiple ways, including carcass theft (kleptoparasitism) and intraguild predation, both of which tend to increase when food is limited (*Palomares and Caro, 1999*; *Donadio and Buskirk, 2006*). Studies of extant carnivores have revealed the significant impact that kleptoparasitism can have on subordinate species that regularly lose their kills to dominant apex predators. As a consequence of calories lost due to kleptoparasitism, European lynx, cheetahs, pumas and wild dogs all have been documented to kill more frequently (*Creel and Creel, 1996*; *Carbone et al., 1999*; *Hayward et al., 2006*; *Krofel et al., 2012*; *Broekhuis et al., 2013*). Moreover, in the case of wild dogs and pumas, kleptoparasitism was associated with consuming carcasses more completely despite the presence of abundant prey (*Carbone et al., 1999*; *Elbroch et al., 2015*). *Carbone et al. (1999)* observed that wild dogs spent more time feeding on carcasses and 'consuming the poorest sections' when spotted hyena (*Crocuta crocuta*) numbers were higher. It is difficult to estimate the added energetic costs incurred by both losing a kill and having to hunt more frequently, but our gray wolf data suggest that the associated need for, or at least advantage of, consuming carcasses more fully results in more rapid wear and risk of tooth fracture, the latter of which can result in debilitating or fatal infections. Thus, rates of tooth wear and fracture in large carnivores can be used as indicators of food limitation and energetic stress load in extant and extinct populations.

For example, Pleistocene guilds of large carnivores were much more species-rich than their present-day equivalents, and often included at least three very large, in some cases social, carnivorous species, such as sabertooth and non-sabertooth cats, large canids, hyaenids, and ursids. It seems likely that kleptoparasitism was a relatively common occurrence, and this would have contributed to the need to utilize carcasses more fully and perhaps scavenge the kills of others, including the consumption of less nutritious and potentially damaging portions, such as bones. Consistent with this, the very high tooth fracture frequencies observed in a number of large Pleistocene carnivores in the Old and New Worlds (*Van Valkenburgh and Hertel, 1993*; *Van Valkenburgh, 2009*; *Flower and Schreve, 2014*) suggest that they experienced 'tough times' or food limitation at more frequent intervals than historic or current populations of similar or the same species. Elevated levels of food stress among apex predators could have had ecosystem wide impacts. Intense food competition

and frequent kleptoparasitism would have favored higher kill rates among subordinate species, and probably an overall greater supply of large carcasses to ecosystems. In addition, it may have been associated with intensified negative effects on herbivore population growth (top-down forcing) as well, but this is difficult to determine without data on fossil herbivore population dynamics.

Finally, our study demonstrates the great value of preserving skeletons, or at least skulls of well-studied mammal populations whenever possible. In our case, dental wear data allowed us to gain insights into the relative difficulty of killing and consuming prey, a critical aspect of large carnivore success that is challenging to quantify in wild populations. Initial studies of tooth fracture in carnivores focused largely on biomechanical questions relevant to tooth strength and consequently, the discovery that tooth fracture can provide insights into levels of food limitation and associated energetic stress was unexpected. There are certain to be more such discoveries based on natural history collections, especially when they are associated with extensive metadata such as those used here (*Schmitt et al., 2018*). Sadly, natural history collections worldwide are under threat due to insufficient funding and a lack of appreciation of their value. Studies such as ours might help reverse this disturbing trend.

## Materials and methods

### Study samples

Isle Royale National Park (IRNP) is an approximately 544 km$^2$ island in Lake Superior within the boundaries of the state of Michigan, USA. The island is inhabited by wolves and moose, both of which have been under intensive study since 1959. There are no other large carnivorans or ungulates on the island and it is essentially isolated from emigration or immigration. Other carnivorans on the island include red fox, pine marten, weasel, mink and river otter, none of which are likely competitors, although foxes are known to scavenge wolf kills. The numbers of wolves and moose are estimated annually from fixed wing aircraft or via ground surveys (*Peterson et al., 2014*). The wolf population has fluctuated, from a maximum of 50 in 1981 to its current minimum of two individuals at the time of this writing (*Peterson et al., 2018*). Over the last 50 years, the ratio of moose to wolf has varied over 8-fold, ranging from 20:1 to 160:1 with an average of 55:1 (*Peterson et al., 2018*). The 64 adult wolf skulls sampled for this paper span 1963–2009 and are housed at Michigan Technological University, Houghton, MI (*Table 1*). Whenever possible, an estimated age at death was recorded for each wolf. Between 1995 and 2008, necropsies were performed on 239 moose killed by wolves, and an estimate of carcass utilization was made that focused largely on the degree of skeletal disarticulation and proportion of bone consumed (*Vucetich et al., 2012*). Over the course of the study, increases in per capita kill rate (#kills/wolf/unit time) were significantly associated with decreases in the proportion of carcass utilized.

The Scandinavian wolf sample consists of 94 skulls collected between 1998 and 2010 and housed in the Swedish Royal Museum of Natural History, Stockholm. Gray wolves were regarded as functionally extinct in Scandinavia in 1966, but began to recolonize south-central Scandinavia in the 1980's, and by 2010, numbered between 250–290 individuals in 52 packs (*Wabakken et al., 2001*). As in IRNP, moose are the primary prey of wolves, but many more calves are taken in southern Scandinavia (Norway, Sweden) than IRNP because they are more abundant due to selective human hunting for bulls and forest management (*Sand et al., 2012*). *Sand et al. (2005)* estimated the proportion of edible biomass (not including bones, rumen, hide and guts) consumed during winter for moose kills made within two wolf territories in southern Scandinavia between 2000–2002.

The Yellowstone National Park sample consists of 160 skulls of adult wolves that are housed in the Yellowstone Heritage and Research Center near Gardiner, Montana. The wolves died between 1996 and 2016, and almost all were from packs inhabiting the northern range of the park. Whenever possible, the age in years and months at death was recorded. Since their initial reintroduction, wolves in this section of the park have been monitored intensively through the application of VHF telemetry and GPS radio-collars, year-round behavioral observations, and genetic analyses as part of the Yellowstone Wolf Project (https://www.nps.gov/yell/learn/nature/wolfreports.htm). Skulls from individuals that were members of the original 31 Canadian founders were excluded from our study as they may have experienced tooth damage while in fenced enclosures that were essential at the start of the project. At the time of wolf reintroduction in 1995, elk numbers in the northern range

exceeded 18,000 and wolves had been absent from the park for over 60 years (*Peterson et al., 2014*). Within two decades after the return of wolves, the number of elk declined to about 5000. Over the same interval, wolf numbers on the northern range rose to exceed 100 but have since declined to between 40 and 50. As a result, the elk:wolf ratio dropped from greater than 600:1 in 1996 to around 100:1 by 2004 and has remained near the lower level for the last 14 years (*Figure 1*). To better understand patterns of wolf predation, Yellowstone Wolf Project personnel completed field necropsies on prey killed by wolves whenever possible. Data collected include the number of skeletal elements present as well as additional data on characteristics of the prey (e.g. species, sex, estimated age, nutritional condition). Data from field necropsies of over 2300 adult elk killed by wolves between 1997 and 2016 were analyzed for this paper to examine whether carcass utilization has changed as elk numbers declined. To explore whether tooth wear and fracture rate shifted over this same interval in Yellowstone wolves, the data were divided into two groups, wolves that died between 1996 and 2006 (n = 74), and those that died between 2007 and 2016 (n = 82). As can be seen from *Figure 1*, the elk:wolf ratio declined rapidly in the first 10 years and then fluctuated around 100:1 for the second decade.

The historic sample of 223 wolf skulls without associated prey data includes individuals collected between 1874 and 1952 from three regions; 1) Alaska, 2) Texas and New Mexico, and 3) Idaho and adjacent Canada (*Table 1*). All these skulls are housed in the National Museum of Natural History, Washington, DC (see *Supplementary file 1*, Table S1).

## Tooth wear and fracture

As in previous studies (*Mann et al., 2017*; *Van Valkenburgh, 1988*; *Van Valkenburgh, 2009*), individual skulls and associated mandibles were examined for dental wear and fracture. To avoid counting teeth that were broken post-mortem or just prior to death due to trap or other damage, teeth were recorded as broken only if there was clear evidence of a fracture (e.g., partially or fully broken cusp) and a blunted surface due to subsequent wear. Missing teeth were not counted as broken, even when alveolar resorption suggested tooth loss due to injury. Consequently, the number of broken teeth are likely underestimates. In addition to recording tooth condition, a qualitative estimate of overall wear stage for the individual was made as follows: 1) 'slight', little or no wear on shear facets or blunting of cusps; 2) 'moderate', shear facets apparent on carnassial teeth and cusps blunted on most teeth; or 3) 'heavy', carnassial teeth with strong shear facets and/or blunted cusps, premolars and molars with well-rounded cusps (*Figure 2—figure supplement 1*).

Tooth fracture incidence was assessed on both a per-individual (percentage of individuals with one or more broken teeth) and a per-tooth (percentage of all teeth that are broken) basis. The distribution of tooth fracture across the tooth row was quantified by tooth type (incisors, canines, premolars, carnassials, post-carnassial molars). Because we were especially interested in detecting the impacts of increased bone consumption, tooth fracture rates were quantified and compared with canine teeth excluded as well as included. Although canine teeth may fracture when feeding on bones, they also may break during intra- and inter-specific combat, as well as when killing prey. Incisors and cheek teeth are used for gnawing and cracking bones, respectively (*Van Valkenburgh, 1996*), and thus are expected to wear more quickly when bone consumption increases. Tooth fracture rates were compared among the study samples using chi-square statistics and the software package SPSS version 25. For the analysis of the effects of age on tooth fracture and wear stage, linear regressions of the number of teeth broken against age for all individuals were performed to test for differences among populations, but results did not achieve significance due to a lack of normality and substantial scatter in the data (*Figure 4—figure supplement 1*). Consequently, comparisons were made among individuals placed into one of three age groups, 1–3 years, 4–6 years, and seven or more years.

## Carcass utilization estimates

In the case of the Scandinavian and Isle Royale samples, carcass utilization data were taken from the literature. In Scandinavia, *Sand et al. (2005)* observed that approximately 70% of the edible biomass of each adult moose kill was consumed by the Scandinavian wolves in their sample, whereas *Vucetich et al. (2012)* estimated that on average, Isle Royale wolves consumed 90% of each moose kill. This difference is greater than it appears given that the Scandinavian estimate of the proportion

of edible biomass consumed did not include bones as potential edible biomass. Had this been done, the proportion consumed would be less than 70%. The relatively low utilization rate in Scandinavia was ascribed to three factors, moose density was high, the moose were naïve to wolf predation, and human disturbance (*Sand et al., 2005*).

For Yellowstone, a skeletal utilization index for over 2300 adult elk kills was constructed based on the percentage of the skeleton that remained after wolves abandoned a kill. The data are derived from the field necropsies mentioned above. At each carcass, observers recorded the number of mandibles, scapulae, vertebrae, humeri, radii plus ulni, femora, tibiae, and presence or absence of metacarpi, metatarsi, skull, and pelvis. The skeletal utilization index used here was calculated as one minus the ratio of the number of mandibles and limb bones found at kill sites to the total number expected if skeletons were complete. Thus if 80% of the skeleton remained, the utilization index was 20%. Data on the status of the skull and pelvis were excluded for this analysis as they were almost always present. For the purposes of this study, it is assumed that missing skeletal elements were most likely removed by wolves, but it is possible that other scavengers, such as bears and coyotes, also may have taken bones. Consequently, the data should be viewed with this in mind. If the skeletal utilization data indicate an increase in bone removal that is not associated with increased tooth wear among wolves, then it might be the case that other carnivores were responsible for the missing bones. However, three-quarters (76%) of the elk necropsies were done during winter when bears are hibernating so their role in carcass utilization is not a factor for the bulk of the sample.

## Acknowledgements

We thank K Cassidy and E Stahler for providing various data for YNP wolves and C Curry for access to and help with the Yellowstone Heritage and Research Center collections. Thanks also to L Vucetich for help with data queries concerning IRNP wolves, and curators at the Swedish Royal Museum of Natural History, Stockholm, the National Museum of Natural History, Washington, DC, and Isle Royale National Park for access to their collections of gray wolves. We thank two anonymous reviewers for suggestions that improved the manuscript. Data collection by BVV was funded in part by US National Science Foundation SGP-1237928. We acknowledge additional funding from US National Science Foundation DEB-1453041 to JAV, Isle Royale National Park (CESU Task Agreement No. P16AC00004, under Master Cooperative Agreement Number P12AC31164), the Robbins Chair in Sustainable Management of the Environment to ROP at Michigan Technological University, and McIntyre-Stennis Grant USDA-NIFA-1014575.

## Additional information

### Funding

| Funder | Grant reference number | Author |
| --- | --- | --- |
| National Science Foundation | SGP-1237928 | Blaire Van Valkenburgh |
| Isle Royale National Park | P16AC00004 | John A Vucetich |
| U.S. Department of Agriculture | NIFA-1014575 | John A Vucetich |
| National Science Foundation | DEB-1245373 | Rolf O Peterson |
| National Science Foundation | DEB-1453041 | Douglas W Smith<br>Daniel R Stahler<br>John A Vucetich |

The funders had no role in study design, data collection and interpretation, or the decision to submit the work for publication.

### Author contributions

Blaire Van Valkenburgh, Conceptualization, Data curation, Formal analysis, Funding acquisition, Investigation, Visualization, Methodology, Writing—original draft, Project administration, Writing—review and editing; Rolf O Peterson, Douglas W Smith, Resources, Data curation, Funding acquisition, Investigation, Methodology, Project administration, Writing—review and editing; Daniel

R Stahler, Data curation, Funding acquisition, Investigation, Methodology, Project administration, Writing—review and editing; John A Vucetich, Resources, Data curation, Funding acquisition, Investigation, Methodology, Writing—review and editing

**Author ORCIDs**
Blaire Van Valkenburgh (iD) https://orcid.org/0000-0002-9935-4719

**Decision letter and Author response**
Decision letter https://doi.org/10.7554/eLife.48628.016

## Additional files

### Supplementary files
• Supplementary file 1. Additional data and figures on wolf tooth wear and fracture, as well as detailed carcass utilization data for Yellowstone National Park wolves.
DOI: https://doi.org/10.7554/eLife.48628.011
• Transparent reporting form
DOI: https://doi.org/10.7554/eLife.48628.012

### Data availability
All data generated or analysed during this study are available at Dryad (https://doi.org/10.5061/dryad.fc8j47n).

The following dataset was generated:

| Author(s) | Year | Dataset title | Dataset URL | Database and Identifier |
| --- | --- | --- | --- | --- |
| Blaire Van Valkenburgh, Rolf O Peterson, Douglas W Smith, Daniel R Stahler, John A Vucetich | 2019 | Data from: Tooth fracture frequency in gray wolves reflects prey availability | https://doi.org/10.5061/dryad.fc8j47n | Dryad Digital Repository, 10.5061/dryad.fc8j47n |

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

1139/z01-029

