## [Decision Letter]

[Editors’ note: minor issues and corrections have not been included, so there is not an accompanying Author response.]

Thank you for submitting your work entitled "Tooth fracture frequency in gray wolves reflects prey availability" for further consideration at *eLife*. Your revised article has been favorably evaluated by Ian Baldwin (Senior Editor), a Reviewing Editor, and two reviewers.

The manuscript was reviewed with favorable comments, though with both reviewers noting some minor corrections to attend to, and offering points to consider in the revision.

Reviewer #1:

In this study, Van Valkenburgh and colleagues present a series of analyses of tooth fracture rates across the dentition using several existing gray wolf skeletal collections with associated metadata. They demonstrate that tooth fracture rates are significantly associated with prey-predator ratios across the samples, even after taking into account potential biases in the age structure and potential per tooth differences in fracture rates. They conclude that tooth fracture frequency is a valid and useful measure of general prey-predator ratio tendencies, and therefore food abundance, in guilds of extinct predators.

The authors integrate gray wolf ecological data from three study areas: Isle Royale, Yellowstone, and Scandinavia, into an impressive "natural experiment" dataset that convincingly demonstrates an association between prey-predator ratios and tooth fracture frequency. The potential biases from age structure and tooth loci within the dentition were accounted for, suggesting that the positive relationship between tooth fracture frequency and prey abundance (or food energetic stress) is a robust observation.

In addition, the manuscript is well-written and easy to read. The plots and tables are useful and necessary to support the discussion and conclusion. The authors might consider taking advantage of the digital online format of the journal to provide color-coded plots to make association of study area data from one plot to the next easier to establish.

Conclusions from this study establish an important baseline for the interpretation of predator-prey dynamics in paleo-ecosystems. The findings from this study are of interest to biologists/ecologists as well as paleobiologists, and is a timely contribution to showcase the importance of well-documented natural history collections for enabling long-term tracking of ecosystem health.

In sum, I have no substantive concerns, and think this manuscript would be ready for publication in *eLife* after minor revisions to address my minor comments.

Reviewer #2:

Extracting behavioral and ecological information from fossils is extremely challenging. The main approach used in the current study of examining tooth fracture rates in carnivores is one such way to assess predator-prey interactions in the fossil record. Despite the vastly greater amount of information that could be obtained from living species through behavioral observations and remote sensing techniques, there are still many aspects of the ecology of extant species that are difficult to ascertain. In the present study, the palaeontological approach is applied to modern populations to convincingly show that tooth fracture rates in large carnivores indicate the degree to which predators are utilizing the carcasses of their prey, and that this likely shows predator population stress due to factors such as lower prey/predator ratios.

This study is a fantastic example of gaining new insight into the ecological dynamics of modern species using palaeontological techniques on museum collections. The additional data about modern populations that can be gained of estimated carcass usage and age structure helps refine the approach and support the conclusions.

The authors rigorously show supporting data for all of their main conclusions by cleverly combining the data from independent studies, each of which could not capture the entire set of variables needed to answer the questions, but can provide support where they do overlap.

I have no substantive concerns about the manuscript.